# Canadian Physicians’ Use of Intramuscular Botulinum Toxin Injections for Shoulder Spasticity: A National Cross-Sectional Survey

**DOI:** 10.3390/toxins15010058

**Published:** 2023-01-10

**Authors:** Farris Kassam, Brendan Lim, Sadia Afroz, Ève Boissonnault, Rajiv Reebye, Heather Finlayson, Paul Winston

**Affiliations:** 1Canadian Advances in Neuro-Orthopedics for Spasticity Congress (CANOSC), Kingston, ON K7K 1Z7, Canada; 2Division of Physical Medicine and Rehabilitation, Université de Montréal, Montreal, QC H3S 2J4, Canada; 3Division of Physical Medicine and Rehabilitation, Faculty of Medicine, University of British Columbia, Vancouver, BC V5Z 2G9, Canada; 4GF Strong Rehabilitation Center, Vancouver, BC V5Z 2G9, Canada

**Keywords:** botulinum toxin, muscle spasticity, spastic hemiplegia, surveys and questionnaires

## Abstract

Spasticity of the upper extremity can result in severe pain, along with many complications that can impair a patient’s activities of daily living. Failure to treat patients with spasticity of the upper limb can result in a decrease in the range of motion of joints and contracture development, leading to further restriction in daily activities. We aimed to investigate the practice patterns of Canadian physicians who utilize Botulinum toxin type-A (BoNT-A) injections in the management of shoulder spasticity. 50 Canadian Physical Medicine and Rehabilitation (PM&R) physicians completed a survey with an estimated completion rate of (36.23%). The demographics of the survey participants came from a variety of provinces, clinical settings, and patient populations. The most common muscle injected for shoulder adduction and internal rotation spasticity was the pectoralis major, this was followed by latissimus dorsi, pectoralis minor, subscapularis and teres major. Injection of BoNT-A for problematic post-stroke shoulder spasticity was common, with (81.48%) of participants responding that it was always or often used in their management of post-stroke spasticity (PSS). Dosing of BoNT-A demonstrated variability for the muscle injected as well as the type of toxin used. The goals of the patients, caregivers, and practitioners were used to help guide the management of these patients. As a result, the practice patterns of Canadian physicians who treat shoulder spasticity are varied, due to numerous patient factors. Future studies are needed to analyze optimal treatment patterns, and the development of algorithms to standardize care.

## 1. Introduction

Spasticity is a sensorimotor disorder characterized by intermittent or sustained involuntary muscle activation, which is a common and potentially problematic consequence of upper motor neuron disorders [1]. While spasticity affecting the shoulder is most studied in stroke patients, shoulder spasticity is seen in other upper motor neuron disorders such as traumatic brain injury, cerebral palsy, spinal cord injury (SCI), and multiple sclerosis (MS). In 2010, among stroke survivors living with spasticity, Wissel et al. reported a prevalence of (58%) of shoulder spasticity at 4 months [2]. Increased muscle overactivity of different combinations of muscles after stroke results in a variety of shoulder posturing patterns (Figure 1) [3]. The shoulder will have an impaired active or passive range of motion due to the increase in tone, or due to musculotendinous retractions which results in muscle shortening and even contracture. Most commonly, increased tone of the pectoralis and subscapularis predominates, resulting in a typical pattern of shoulder internal rotation and adduction [4,5]. Of the 5 cardinal positions described by Hefter et al., position 3, with the shoulder internally rotated and adducted, the elbow flexed, and the wrist and forearm in neutral were found to be most common in stroke [6].

Spasticity of the upper extremity leads to many complications, including, pain, and the impairment of patients’ activities of daily living such as hygiene and dressing. Failure to treat spasticity of the upper limb can result in a loss of joint range of motion and contracture development, which can then further exacerbate participation restriction [7]. 

While the literature often focuses on the pectoralis major and subscapularis, there are many muscles implicated in shoulder spasticity including the deltoid, trapezius, teres major and minor, pectoralis major, subscapularis, supra- and infra- spinatus, coracobrachialis, latissimus dorsi and the long head of the biceps and triceps brachii muscles [8]. The choice of muscles to inject is largely due to personal bias or experience. A 2022 European consensus identified the expert groups’ preferences. This included the teres major and deltoid muscles as frequent targets in addition to the commonly studied subscapularis and pectoralis major muscles [8].

Botulinum Toxin type A (BoNT-A) can be used for the treatment of focal spasticity [9]. Canadian Stroke Best Practice Guidelines state that the use of BoNT-A for upper limb spasticity to increase range of motion and decrease pain is supported by Level B evidence in less than 6 months post-stroke and Level A evidence more than 6 months post-stroke [10]. Multiple randomized controlled trials have studied the use of BoNT-A in the shoulder girdle muscles for post-stroke spasticity [11,12,13,14] and there is conflicting evidence on its efficacy, likely related to variability in muscles, doses, and dilutions injected. There are currently no publications that describe how Canadian physicians currently use BoNT-A to treat spasticity of the shoulder muscles.

We aimed to determine the current practices of Canadian physicians who commonly manage shoulder girdle spasticity. We created an online survey to determine the current treatment techniques, patient outcomes, complications, and barriers to treatment of shoulder spasticity. 

## 2. Results

### 2.1. Participant Criteria

Participants in this study were licensed Canadian physicians who use BoNT-A for spasticity management; resident physicians or medical students were excluded. 79 physician respondents opened our survey, 21 respondents submitted partially completed questionnaires and 4 respondents did not use BoNT-A for spasticity management, and as a result, both groups were therefore excluded from the study. Finally, 54 completed questionnaires were included in the final analysis. Approximately 138 Canadian PM&R physicians involved in spasticity management have been reflected in previous studies by Kassam et al. that also evaluated practice patterns of Canadian physicians involved in spasticity [15]. Of the 54 respondents included in the final analysis, 50 were PM&R physicians, giving an approximate response rate of (36.23%) of Canadian PM&R physicians involved in spasticity management (Figure 2). Completed questionnaires were defined when the participant answered at least (50%) of all questions in a skip logic setting.

### 2.2. Clinicians’ Demographics

This study included physicians who practiced in 7 provinces, and (90.74%) were exclusively PM&R specialists. All participants (*n* = 54) were experienced in the use of intramuscular BoNT-A with (55.56%) having used BoNT-A for more than 10 years. The physicians worked in multiple settings and across multiple patient populations. (55.56%) worked in acute care, (74.07%) worked in rehabilitation units, and (88.89%) worked in the outpatient setting (Table 1). (63.67%) of participants treated adults exclusively, and (31.48%) treated adults and children. The majority of participants treated stroke (96.30%), multiple sclerosis (83.33%), cerebral palsy (81.48%), spinal cord injury (79.63%), and traumatic brain injury (77.78%) (Table 2).

### 2.3. Patient Profiles and Interventions

Using Hefter et. al’s classification system (Figure 1), upper limb spasticity patterns from all etiologies were ranked in the following order from most to least common: patterns 4 > 1 > 3 > 5 > 2 (Figure 3). Post-stroke spasticity had a significantly higher likelihood of manifesting shoulder adduction and internal rotation when compared to the other causes of upper limb spasticity (Table 2). Similarly, problematic spasticity was more common in stroke patients in comparison to the other causes of upper limb spasticity (Table 2). If contracture was suspected to be the cause of shoulder internal rotation or adduction, physicians rarely referred patients for surgical release (Table 2).

For patients presenting with shoulder adduction and internal rotation, pectoralis major was noted as the most frequent muscle treated with BoNT-A. This was followed by latissimus dorsi, pectoralis minor and subscapularis that were treated in similar frequencies. Teres major was ranked as the muscle least likely to be treated with BoNT-A (Figure 4). A majority of practitioners (79.63%) always or often considered injection with BoNT-A in the muscles of the shoulder girdle in their first round of management when treating upper limb spasticity (Table 3). For most practitioners (92.31%), the minimum threshold for BoNT-A injection into the shoulder muscles was at least a Modified Ashworth Scale (MAS) score of +1 (Table 3).

### 2.4. Intervention Techniques and Timing

When BoNT-A was injected into the shoulder girdle, electromyography (78.84%) was the most frequent localization method, followed by ultrasound (59.62%), electrical stimulation (48.08%) and lastly anatomical landmarks (32.69%) (Table 3). A majority (88.46%) of physicians grasped the muscle belly of the pectoralis major or latissimus dorsi when injecting BoNT-A (Table 3). A majority (79.63%) of physicians often or always injected BoNT-A in the first round of shoulder spasticity management if the patients presented with adduction and/or internal rotation.

### 2.5. Dosing

The dosing of toxin varied for each muscle and type of BoNT-A used. Participants were asked for a dose range of toxin used for each muscle, thus typical descriptive statistics could not be calculated. We report a great variability in the dosage used to inject each muscle. (Table 4), shows the minimum and maximum dose used for each muscle and type of BoNT-A. In addition, only 3 participants (5.77%) use phenol injections to treat shoulder adduction or internal rotation (Table 3).

### 2.6. Patient Outcomes

Participants strongly agreed that the use of BoNT-A to treat upper limb spasticity was beneficial with minimal complications (Table 5, Table 6 and Table 7). Physician use of BoNT-A to treat shoulder girdle spasticity included an improved range of motion, increased function, pain reduction, decreased skin breakdown, increased participation in activities of daily living, reduced caregiver burden, improved hygiene and/or meeting the goals set in collaboration with the patient and their caregivers (Table 6). Participants also noted minimal side effects that included mild and short-term localized pain and discomfort, along with weakness (Table 7). Additionally, two physicians noted a total of 3 incidents of pneumothorax over their span of practice to date (Table 3).

### 2.7. Barriers to Treatment

A total of (61.54%) of physicians noted that there were barriers in administering intramuscular BoNT-A for patients with shoulder spasticity as compared to other upper extremity areas. Clinic capabilities or constraints were noted by participants as being the main barriers to treatment. These included physician/clinic financial resources (15.38%), clinics not equipped with the necessary equipment (13.46%), clinician time constraints (9.62%), and lack of interdisciplinary care (26.92%)**.** None (0%) of the respondents felt that there was a lack of effectiveness of BoNT-A in the clinical setting (Table 8).

## 3. Discussion

The primary finding of this study was that Canadian physicians use BoNT-A as a common intervention to treat shoulder spasticity, despite the absence of official indication for shoulder muscles on the BoNT-A product monographs. This is a similar finding to Holmes et al.’s study that investigated the treatment patterns of physicians in the United Kingdom and Carvalho et al. from Portugal [5,16]. Our study sampled a large and diverse physician group with respect to geographic area, patient populations, clinic location, and clinical experience using BoNT-A. In Canada, PM&R leads the practice of non-surgical interventions for the treatment of spasticity. By enlisting participants through CANOSC, the largest national organization of mainly PM&R physicians that specialize in the treatment of spasticity, the physicians of this study most likely provide non-surgical interventions to the majority of spastic patients in the entire country. 

Our study is novel as it included spasticity from multiple etiologies and not just post-stroke. It additionally assessed the patient and physician-specific factors for initiating treatment including goals of treating spasticity, how commonly the shoulder was treated, the units utilized for each muscle of three available BoNT-A preparations, complications of BoNT-A, method of guidance for injections, as well of barriers to treatment with BoNT-A. We also assessed the use of alternative therapies such as phenol and surgery.

The use of intramuscular BoNT-A to treat shoulder spasticity is highly variable among physicians and clinics. Variability is to be somewhat expected based on the complexity of the patient, which includes the number of muscles involved, degree of spasticity, level of disability, the experience of the practitioner, and patient goals. Without an established treatment algorithm, the course of treatment is ultimately left to the discretion of the physician and their preferences, their ability to consult with complementary practitioners, accessibility to equipment, physician preference in muscle selection, preference in injection techniques and modalities used, and determining the minimum criteria needed for intervention, all resulting in differences in treatment patterns.

The muscle most commonly injected for the treatment of shoulder spasticity was pectoralis major, which was also found in previous studies [2,16,17]. While the pectoralis was the most common muscle injected across our study and others, the remaining muscles were highly varied. For example, our participants injected subscapularis and latissimus dorsi in similar frequencies, while Nalysnk et al. found that the subscapularis was injected 1.9 times more than latissimus dorsi. Emerging RCTs have shown that BoNT-A injections into the subscapularis have shown improvements in spasticity, range of motion and pain of the shoulder [13]. The reason for this variability is potentially related to the on-label indications of the individual BoNT-A product monographs where the shoulder girdle muscles are not indicated. 

Hefter et al.’s classification system is one of the first attempts to classify common patterns of upper limb spasticity exclusively due to stroke and outlines five upper limb postures that captured (94%) of patients post-stroke [3]. Further studies have reconfirmed the original findings, showing near unanimous distribution into the 5 post-stroke postures [18,19] and with position 3 (shoulder internal rotation and adduction, elbow flexion, and neutral forearm and wrist) being the most common in all of the studies. While these findings are favourable for establishing Hefter et al.’s system to be used in practice for stroke patients, Gomes et al. noted that Hefter et al.’s classification system has its limits in practice as evaluators could reach an agreement in only (67%) of cases in the first round of independent classifications (3,18).

Participants in this study ranked the patterns they see most often from all causes of upper limb spasticity, as opposed to those exclusively found in stroke patients. With the inclusion of all pathologies, all five of Hefter et al.’s positions were still noted to be present in clinical practice. While all 5 patterns were observed by participants, the frequency of upper limb spasticity patterns was ranked as 4 > 1 > 3 > 5 > 2. This is slightly different from the frequency reported by Hefter et al. in their original paper, in which upper limb spasticity patterns were ranked as 3 > 1 > 4 > 2 > 5. We recognize the limitations of a comparison between prospective measurement and retrospective clinician memory, which is susceptible to bias. Therefore, future studies are needed to further explore the potential validity of Hefter et al.’s classification system to all causes of upper limb spasticity. Establishing a validated classification system could lead to earlier training in upper limb spasticity management in medical training, a reduction in physician time constraints, which is noted as a major barrier to the treatment of upper limb spasticity (Table 8), and more precise muscle selection, ultimately leading to the optimization of patient care and quality of life.

A recent 2022 expert consensus by Jacinto et al. adapted the five post-stroke spasticity patterns into two shoulder spasticity patterns: “A” (adduction, elevation, flexion and internal rotation of shoulder) with the presence of elevation and flexion, vs. extension in Pattern “B” (abduction or adduction, extension and internal rotation of the shoulder. It is notable that Jacinto et al.’s expert panel made further suggestions to define patterns [8].

Variability between physicians also existed for the extent of spasticity before treatment with BoNT-A was considered. The MAS is one of the most common methods of assessing the degree of spasticity and can be applied to different joints, and for different populations affected by tone [20]. (65.38%) of Canadian physicians considered treatment at a MAS value of +1 or less, possibly indicating Canadian physicians may provide earlier intervention in comparison to their international colleagues. Further studies could explore other off-label utilization of BoNT-A in the hemiplegic upper extremities, such as pain and spasticity prevention in patients with significant risk factors of developing spasticity. Interestingly, Wissel et al. required a minimum MAS rating of 2 to be included in a study that showed the beneficial effect of BoNT-A on the shoulders of patients with spasticity [21].

EMG was the most frequently (78.84%) used localization technique for targeting muscles with BoNT-A. Since EMG was the first instrument to help with the guidance of these injections, this finding was expected. Ultrasound is also becoming more recognized as an effective tool [22] and it was the next most commonly used modality, with (59.62%) of participants using it around the shoulder. Studies have shown that the use of instrumented guided BoNT-A injections has improved clinical results compared to landmarks alone [23], and yet, (32.69%) of physicians have had a situation where only anatomical landmarking was used. As a result, an analysis of the potential barriers to using instrument-guided modalities is needed to reduce the number of Canadian practitioners who inject BoNT-A without any instruments. 

With the introduction of new equipment and techniques comes a financial cost, as well as further training needed to operate. (13.46%) of Canadian participants stated that their clinic did not have the necessary equipment to treat patients with shoulder spasticity with BoNT-A. Patterns of localization techniques could change in the future, as equipment costs inevitably decrease, and the training for physicians who treat spasticity become more reliant on ultrasound, likely a testament to their preference of treatments. In addition, Liang et al. noted that spasticity can often be underrecognized and present alongside numerous accompanying issues [24]. If upper limb classification systems such as Hefter et al.’s framework, and treatment algorithms were established, this would ultimately encourage changes in treatment practices, minimize variability in care as physicians would not be targeting muscles based only on personal preferences, and improve patient outcomes. 

Canadian physicians felt that BoNT-A was an effective non-surgical intervention for patients with shoulder spasticity as it reduced pain, improved range of motion, improved participation in activities of daily living and decreased caregiver burden **(Table 6**). None of the Canadian physicians felt that BoNT-A was ineffective as an intervention in the clinical setting (Table 8). As with any intervention, there is always a degree of risk. However, Canadian physicians found temporary minor complications with (94.23%) of participants never having a single pneumothorax over the span of their career (Table 3). This sentiment is strongly reinforced by Choi et al., Johst et al. and Wissel et al. [19,21,25].

Finally, participants noted a wide range of doses (Table 4) when administering Onabutulinum Toxin A, Abobotulinum Toxin A or Incobotulinum Toxin A. This is to be expected as the amount of BoNT-A needs to be tailored based on the degree of spasticity, the patient’s goals and the optimization of function. In line with the previous studies [26], Canadian physicians rarely use phenol (5.77%) to treat shoulder spasticity. However, Li et al. found that if post-stroke spastic patients were administered phenol first, the total amount of BoNT-A used to manage patients was reduced without an increase in side effects [27]. Future investigations in upper limb spasticity could aim to determine if the integration of phenol and BoNT-A could maximize patient outcomes.

Limitations of this study include voluntary response bias, where participants with an increased interest in the study’s focus increase the likelihood of participation. This may lead to an under-representation of the targeted study population. There is also the possibility of response bias where participants select responses that are more in line with the expected practice rather than their clinical practices, which we attempted to minimize by ensuring that this survey remained anonymous. We also did not examine treatment outcomes from the patient or allied health professional perspective. While this study had a small sample size, it likely elucidates the practices of Canadian PM&R specialists involved in shoulder spasticity in both inpatient and outpatient settings. Furthermore, this is the largest study to date examining Canadian PM&R specialists’ treatment practices for shoulder spasticity. Finally, because of the snowball sampling methodology, we cannot trace how many people received the link, and consequently, we were unable to calculate a precise response rate. 

## 4. Conclusions

Our study revealed that Canadian participants use BoNT-A as a beneficial treatment for shoulder spasticity with minimal risks and complications. Without the development of classification systems and treatment algorithms for shoulder spasticity, variability in treatment will continue to be based on physician preferences. Future directions include the creation of treatment algorithms for shoulder spasticity to help patients receive uniformity of care no matter where they reside in Canada.

## 5. Materials and Methods

### 5.1. Study Design

A cross-sectional survey of Canadian physicians with experience in spasticity management was designed and administered via an online platform. Research ethics approval was obtained by the local institutional research ethics board.

### 5.2. Survey Design

Participants were invited to complete the survey through the web-based platform, Alchemer (www.alchemer.com, Louisville, CO, USA). All data were collected electronically. The CAPMR (Canadian Association of Physical Medicine and Rehabilitation) secretariat maintains an enterprise license for this software. 

The survey was developed with the input of the 3 senior authors [(PW, HF, RR)]. The survey questions were further refined with the input of other Physical Medicine & Rehabilitation (PM&R) physicians who treat spasticity and medical students. The questionnaire employed a mix of multiple choice, Likert scale, rank order, and open-ended questions. A question skip logic was applied. For example, if participants selected that they do not inject BoNT-A for shoulder adduction spasticity, then the survey skipped the follow-up question asking which muscles participants inject. The final version comprised 50 total questions. This included physician demographics, patient profiles, the prevalence of the patterns of shoulder spasticity, muscles targeted, intervention techniques, patient outcomes, treatment complications, barriers to treatment, and participant insight into their clinical experience. Survey questions sought both quantitative and qualitative responses. While the survey had several questions on pain management and the timing of treatments, these findings will be used to design future studies on hemiplegic pain management. The full survey is Appendix A in manuscript. For this paper, we focused exclusively on the first 36 questions. 

### 5.3. CANOSC

Canadian Advances in Neuro-Orthopedics for Spasticity Consortium (CANOSC) is a federally incorporated organization under CAPMR and is Canada’s largest organization dedicated to enhancing the field of spasticity through education and collaboration in an interdisciplinary setting. While a majority of its 167 members are PM&R specialists, members also include plastic surgeons, orthopedic surgeons, pediatricians, neurologists, and neurosurgeons, along with allied health professionals including physiotherapists and occupational therapists. A 2019 survey conducted by the Canadian Medical Association estimated there were 500 Canadian PM&R specialists in clinical practice [28]. CANOSC comprises 113 Canadian Physiatrists, approximately (22.6%) of all active PM&R physicians in Canada, and a majority of all Canadian PM&R physicians who treat spasticity. These demographics make it an ideal organization to examine upper limb spasticity treatment practices in Canada.

### 5.4. Participant Recruitment

The sample population for this study was Canadian physicians involved in the management of shoulder spasticity. An invitation email was distributed to the email list of CANOSC using an online link. It contained information about our study, obtaining consent, privacy information, ethics statements, and the link to the anonymized survey platform. Snowball sampling was also employed as recipients of the email were asked to forward the link to other appropriate practitioners. A follow-up reminder email was sent 6 weeks later. The online survey tool did not record any identifying information, to ensure the anonymity of responses. No offers of incentives were offered to participants. Upon submission, all questionnaires were thoroughly reviewed by two of the study authors and a data analyst, and any disagreements were resolved through consensus. Partially responded surveys, defined by completion of less than (50%) out of the 36 questions relevant to this paper, were excluded. The descriptive data from the completed surveys were extracted. The survey platform was open between August 2021 and November 2021. 

### 5.5. Data Analysis

This study collected primary data and securely stored it on the web-based platform, *Alchemer*. Data were then downloaded into an Excel spreadsheet and imported into the R statistical software package for data analysis. Descriptive statistics were analyzed to describe, synthesize, and summarize the data. Content analysis was performed on open-ended responses to report the findings. 

## Figures and Tables

**Figure 1 toxins-15-00058-f001:**
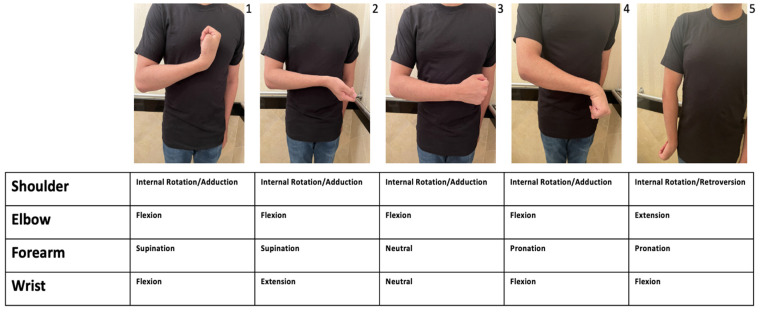
Hefter et al.’s [3] classification of spastic upper extremity patterns.

**Figure 2 toxins-15-00058-f002:**
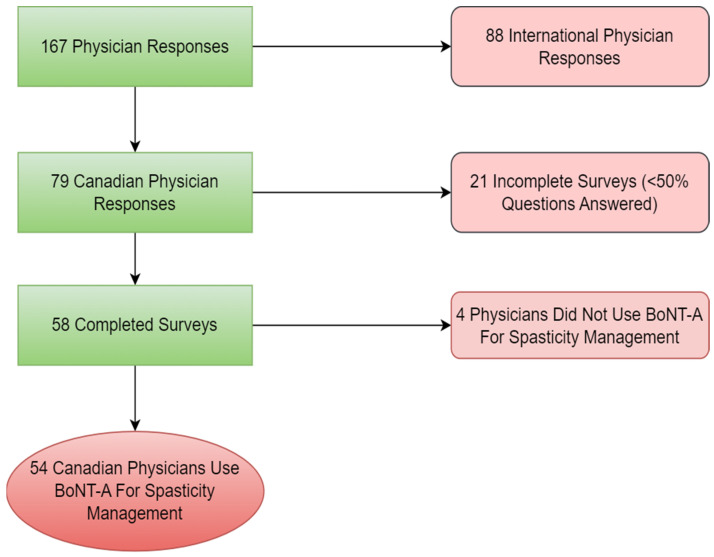
Flow chart of partial and complete survey participants.

**Figure 3 toxins-15-00058-f003:**
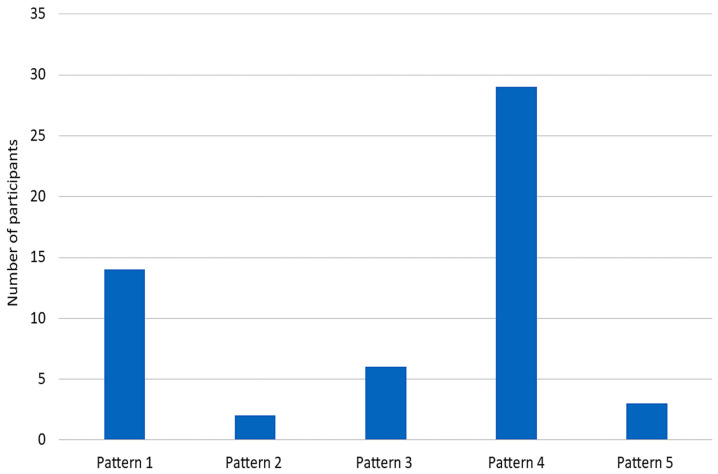
Frequency of upper limb spasticity patterns (Ranked 1st).

**Figure 4 toxins-15-00058-f004:**
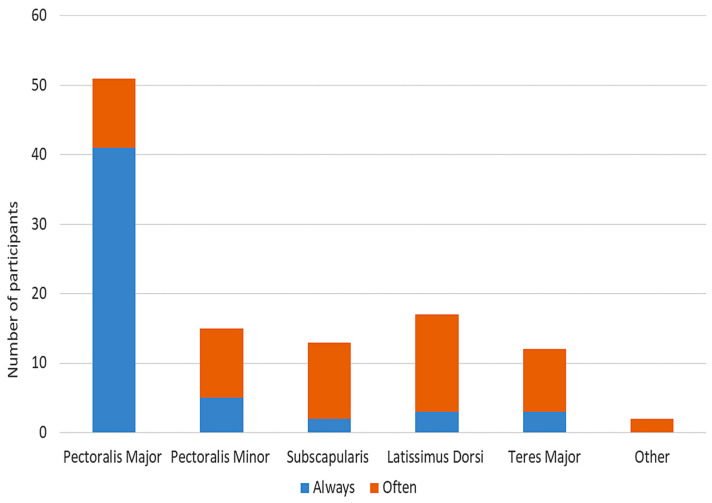
Frequency of injecting BoNT-A for shoulder adduction and internal rotation.

**Table 1 toxins-15-00058-t001:** Physician demographics.

		% (*n*)
Location	British Columbia Ontario Quebec Alberta Saskatchewan Newfoundland and Labrador New Brunswick	33.33% (18) 24.07% (13) 20.37% (11) 12.96% (7) 3.70% (2) 3.70% (2) 1.85% (1)
Speciality	PM&R Neurology PM&R and Pediatrics	90.74% (49) 7.41% (4) 1.85% (1)
Setting for Spasticity Assessment and Management	Outpatient clinic Rehabilitation unit Acute care setting Other	88.89% (48) 74.07% (40) 55.56% (30) 7.41% (4)
Number of Years Injecting BoNT-A	0–9 10–19 20–29 >30	44.44% (24) 37.04% (20) 14.81% (8) 3.6% (2)

**Table 2 toxins-15-00058-t002:** Patient demographics.

		% (*n*)
Patient’s Treated	Adults only Both children and adults Children only	63.67% (36) 31.48% (17) 1.85% (1)
Service provided to the spasticity population of:	Stroke	96.30% (52)
	Cerebral Palsy	83.33% (45)
	Spinal Cord Injury	81.48% (44)
	Traumatic Brain Injury	79.63% (43)
	Multiple Sclerosis	77.78% (42)
	Other and idiopathic (e.g., Hereditary spastic paraparesis, ALS)	77.07% (40)
When considering post stroke spasticity (PSS), how often do you observe that the shoulder is held in the internally rotated and adducted position?	50–100% 0–50%	92.59% (50) 7.41% (4)
When considering other causes of spasticity, including MS, SCI, and CP, how often do you observe that the shoulder is held in the internally rotated and adducted position?	50–100% 0–50%	53.70% (29) 46.29% (25)
When considering upper extremity PSS, how often do you identify problematic spasticity that requires management with BoNT-A in the shoulder as part of your plan?	Always/Often Sometimes Seldom	81.48% (44) 16.67% (9) 1.85% (1)
When considering other causes of upper limb spasticity with problematic adduction and internal rotation, including MS, SCI, and CP, how often do you identify problematic spasticity that requires management with BoNT-A in the shoulder as part of your plan?	Always/Often Sometimes Seldom Never	46.29% (25) 38.89% (21) 12.96% (7) 1.85% (1)
If you suspect contracture causing shoulder internal rotation and/or adduction, have you referred such patients for surgical release?	Always Often Sometimes Seldom Never Not Applicable	1.92% (1) 9.62% (5) 15.38% (8) 15.38% (8) 44.23% (23) 13.46% (7)

**Table 3 toxins-15-00058-t003:** Physician practice patterns.

		% (*n*)
Do you inject the muscles of the shoulder girdle with BoNT-A for spasticity management?	Yes	100% (54)
How often do you include shoulder muscles in your first round of management with BoNT-A if adduction and/or internal rotation spasticity is identified?	Always Often Sometimes Seldom	25.93% (14) 53.70% (29) 16.67% (9) 3.70% (2)
What is the minimum Modified Ashworth Scale you will inject BoNT-A for the shoulder?	0 1 1+ 2 3	1.92% (1) 5.77% (3) 57.69% (30) 23.08% (12) 11.54% (6)
What target muscle localization methods do you use for BoNT-A injection around the shoulder girdle?	Electromyography Ultrasound Electrical stimulation Anatomical landmarks only Other	78.84% (41) 59.62% (31) 48.08% (25) 32.69% (17) 1.92% (1)
Do you ever grasp the wad (muscle belly) of the pectoralis muscles or latisimus dorsi in your hand when you inject to avoid going too deep?	Yes No	88.46% (46) 11.53% (6)
List the number of lung punctures that have occurred with shoulder muscle injections that caused a pneumothorax for you personally?	0 1 2	94.23% (49) 3.85% (2) 1.92% (1)
Do you use phenol or alcohol for shoulder adduction or internal rotation spasticity?	Yes No	5.77% (3) 94.23% (49)
Which nerves do you target?	Lateral Pectoral Nerve Medial Pectoral Nerve Subscapular Nerve	66.67% (2) 66.67% (2) 33.33% (1)

**Table 4 toxins-15-00058-t004:** Dosing ranges of BoNT-A subtypes by muscle.

	Botox Onabutulinum Toxin A	Dysport Abobotulinum Toxin A	Xeomin Incobotulinum Toxin A
Pectoralis Major	10–200	50–500	25–200
Pectoralis Minor	10–100	50–150	10–75
Subscapularis	15–150	50–400	25–150
Latissimus Dorsi	15–200	40–500	25–200
Teres Major	10–100	50–300	20–100
Deltoid	10–200	50–300	20–200

**Table 5 toxins-15-00058-t005:** Goals for BoNT-A intervention.

		% (*n*)
When you inject the shoulder muscles with BoNT-A for adduction and internal rotation, what are your goals?	Achieve a patient or caregiver goal such as using the Goal Attainment Scale	96.15% (50)
	Reduce pain	94.23% (49)
	Increase range of motion	86.53% (45)
	Reduce spasticity as measured with the Modified Ashworth Scale or similar	44.23% (23)
	Other	9.6% (5)

**Table 6 toxins-15-00058-t006:** Benefits of BoNT-A intervention.

What Improved Outcomes Have You Observed to the Use of BoNT-A Injections on Patients with Shoulder Spasticity?
Improved pain, ROM and function of shoulder and arm.
Reduced pain, improved function, increased comfort, facilitated cares & reduced skin breakdown.
Better ease of dressing and hygiene.
Reduction in pain, increased range of motion active and passively & easier care for caregivers.
Increase ROM decrease pain ease motion for caregivers.
Able to dress self, decreased caregiver burden & less pain.
Pain reduction, improved passive participation in ADL & improved caregiver satisfaction.
Lots. All goals set in collaboration with patient/caregiver. Active and passive.

**Table 7 toxins-15-00058-t007:** Complications for BoNT-A intervention.

What Complications Have You Observed with the Use of BoNT-A Injections on Patients with Shoulder Spasticity?
None
Increased short term discomfort
Minimal
Weakness with reduced function

**Table 8 toxins-15-00058-t008:** Barriers to BoNT-A intervention.

		% (*n*)
What are barriers to the use of BoNT-A for patients with shoulder spasticity as compared to other upper extremity regions?	No barriers	38.46% (20)
	Financial—patient resources	36.54% (19)
	Lack of interdisciplinary care	26.92% (14)
	Patient does not want to have BoNT-A treatment	17.31% (9)
	Financial—physician/clinic resources	15.38% (8)
	Clinic not equipped with necessary equipment	13.46% (7)
	Risk of adverse events	11.54% (6)
	Clinician time constraints (e.g., clinical practice too busy)	9.62% (5)
	Other	9.62% (5)
	Off-Label	5.77% (3)
	Lack of evidence in the literature	3.85% (2)
	Lack of effectiveness from clinical experience	0% (0)

## Data Availability

All data will be kept on a secure password protected file for 15 years.

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
