# Peer review of "Canadian Physicians’ Use of Intramuscular Botulinum Toxin Injections for Shoulder Spasticity: A National Cross-Sectional Survey"

_toxins, 2023, doi:10.3390/toxins15010058_

Round 1
Reviewer 1 Report
the research is intriguing and provides a thoughtful point for discussion of the issue of shoulder spasticity.
please consider the following suggestions to improve your paper.
Minor critiques and suggestions
Introduction
Introduction section should be improved. Authors should add a better definition of shoulder spasticity adding details about structural and functional anatomy of the muscle involved and the impact on both active and passive functions.
- figure 1 representing the "most prevalent .."is of a low quality (fish-eye effect) and it is not necessary to improve readability of the article. Please consider to remove it.
- Authors should clarify the novelty of the study.
Results
- table 3: in some row the sum of participants answers is 52 instead of 54 (total number of participants included). Please verify.
- table 6 and table 7 do not report participations answers. Authors should add these data.
Discussion
- hefter pattern classification is a method to differentiate abnormal posture in patients with arm spasticity by means of clinical observation. It is too ambitious to express numerical data based on a survey and on clinician's memory. It is not methodologically correct a comparison between frequency data from literature and an expert opinion from a survey. I strongly suggest the authors to rewrite lines 260-271.
- I would recommended referencing:
Carvalho MP et al Top stroke rehabilitation 2018 "Analysis of 15-years' experience in including shoulder muscle, when treating upper limb spasticity with botulinum toxin type A.
-In a recent study there were identified two patterns of shouklder spasticity: Pattern A and Pattern B. Authours shoul add this reference and a comment.
Please see the following reference: Jacinto J et al. 2022 Front. Neurol. "A practical guide to botulinum neurotoxin treatment of shoulder spasticity: anatomy, physiology and goal setting"
- please write intramuscular instead of IM at line 232 pg 10
Author Response
Our Responses to our reviewers:
Our team would like to thank you for prompts and excellent suggestions. We have reviewed all suggestions and thank you for improving our scholarship.
Minor critiques and suggestions
Introduction
Introduction section should be improved. Authors should add a better definition of shoulder spasticity adding details about structural and functional anatomy of the muscle involved and the impact on both active and passive functions.
To address this suggestion, we have added in the definition by Pandyan. We have explained spasticity and muscle shortening. We note that in all of the papers reviewed the mechanism of action of each muscle implicated in spasticity is not routinely discussed. There are 18 muscles that attach to the scapula. We have added in details from Jacinto et al, 2022: “There are many muscles implicated in the shoulder spasticity including: the deltoid, trapezius, teres major and minor, pectoralis major, subscapularis, supra- and infra- spinatus, coracobrachialis, latissimus dorsi and the long head of the biceps and triceps brachii muscles” (Jacinto, Camões-Barbosa, Carda, Hoad, & Wissel, 2022). It is not possible to add in the mechanisms of all of these muscles.
- figure 1 representing the "most prevalent .."is of a low quality (fish-eye effect) and it is not necessary to improve readability of the article. Please consider to remove it.
We have modified the description as ‘Hefter et al.’s classification of spastic upper extremity patterns.’
- Authors should clarify the novelty of the study.
We believe that our study is novel as it has included spasticity from multiple etiologies and not just post-stroke. Our study additionally assessed the patient and physician specific factors for initiating treatment including: goals of treating spasticity, how commonly the shoulder was treated, the units utilized for each muscle of three available BoNT-A preparations, complications of BoNT-A, method of guidance for injections, as well of barriers to treatment with BoNT-A. We also assessed the use of alternative therapies such as phenol and surgery. By examining multiple etiologies of shoulder spasticity, our hope is that through publication, we can progress the field of treating spasticity of the shoulder, ultimately improving patient’s quality of life. We also recognize that our paper does have limitations, which are also discussed.
Results
- table 3: in some row the sum of participants answers is 52 instead of 54 (total number of participants included). Please verify.
This is verified. We included survey responses where over 50% of the questions were answered. For 2 participants, they declined to respond to later questions, but still answered over 50%, thus they are included in our data set.
- table 6 and table 7 do not report participations answers. Authors should add these data.
Tables 6 and 7 sample from participants answers to highlight the improved outcomes and complications that have been clinically found.
Discussion
- Hefter pattern classification is a method to differentiate abnormal posture in patients with arm spasticity by means of clinical observation. It is too ambitious to express numerical data based on a survey and on clinician's memory. It is not methodologically correct a comparison between frequency data from literature and an expert opinion from a survey. I strongly suggest the authors to rewrite lines 260-271.
We agree with this suggestion and we have not taken any conclusions. Our intention is to point out that individual physician practices and different pathologies have variation.
- I would recommended referencing:
Carvalho MP et al Top stroke rehabilitation 2018 "Analysis of 15-years' experience in including shoulder muscle, when treating upper limb spasticity with botulinum toxin type A.
Thank you, we have included this.
-In a recent study there were identified two patterns of shoulder spasticity: Pattern A and Pattern B. Authours should add this reference and a comment.
Please see the following reference: Jacinto J et al. 2022 Front. Neurol. "A practical guide to botulinum neurotoxin treatment of shoulder spasticity: anatomy, physiology and goal setting"
We have included this reference, thank you.
- please write intramuscular instead of IM at line 232 pg 10
We have completed this.
Our team would like to thank the reviewers for their time and efforts to improve our work.
Reviewer 2 Report
I enjoyed reading this paper. It is always interesting to see how people are doing things, why that might be and what can be improved. I agree with the points made in your discussion about future directions. I do wonder whether the respondents were the main injectors and that those that did not respond did not respond because, for whatever reason, they do not inject the shoulder. I also wonder whether there are any differences in the practices for children ( with CP/dystonia) and adults.
It is a well written, clear paper which delivers on its aims.
Author Response
Thank you for your comments.